# Predictive Values of Location and Volumetric MRI Injury Patterns for Neurodevelopmental Outcomes in Hypoxic-Ischemic Encephalopathy Neonates

**DOI:** 10.3390/brainsci10120991

**Published:** 2020-12-16

**Authors:** Peter D. Chang, Daniel S. Chow, Anna Alber, Yen-Kuang Lin, Young Ah Youn

**Affiliations:** 1Department of Radiological Sciences, Center for Artificial Intelligence in Diagnostic Medicine (CADIM), University of California, Irvine, CA 92697, USA; changp6@hs.uci.edu (P.D.C.); chowd3@hs.uci.edu (D.S.C.); aalber@hs.uci.edu (A.A.); 2Research Center of Biostatistics, Taipei Medical University, Taipei City 106, Taiwan; robbinlin@tmu.edu.tw; 3Department of Pediatrics, Seoul St. Mary’s Hospital, College of Medicine, The Catholic University of Korea, Seoul 06591, Korea

**Keywords:** diffusion weighted MRI, hypothermia, hypoxia-ischemia, seizures, long term outcomes

## Abstract

Hypoxic-ischemic encephalopathy (HIE) is a severe neonatal complication with up to 40–60% long-term morbidity. This study evaluates the distribution and burden of MRI changes as a prognostic indicator of neurodevelopmental (ND) outcomes at 18–24 months in HIE infants who were treated with therapeutic hypothermia (TH). Term or late preterm infants who were treated with TH for HIE were analyzed between June 2012 and March 2016. Brain MRI scans were obtained from 107 TH treated infants. For each infant, diffusion weighted brain image (DWI) sequences from a 3T Siemens scanner were obtained for analysis. Of the 107 infants, 36 of the 107 infants (33.6%) had normal brain MR images, and 71 of the 107 infants (66.4%) had abnormal MRI findings. The number of clinical seizures was significantly higher in the abnormal MRI group (*p* < 0.001) than in the normal MRI group. At 18–24 months, 76 of the 107 infants (70.0%) showed normal ND stages, and 31 of the 107 infants (29.0%) exhibited abnormal ND stages. A lesion size count >500 was significantly associated with abnormal ND. Similarly, the total lesion count was larger in the abnormal ND group (14.16 vs. 5.29). More lesions in the basal ganglia (BG) and thalamus areas and a trend towards more abnormal MRI scans were significantly associated with abnormal ND at 18–24 months. In addition to clinical seizure, a larger total lesion count and lesion size as well as lesion involvement of the basal ganglia and thalamus were significantly associated with abnormal neurodevelopment at 18–24 months.

## 1. Introduction

Hypoxic-ischemic encephalopathy (HIE) after perinatal asphyxia affects 1–8 per 1000 live births and results in 15~25% mortality; among infants who survive, 25% have permanent neurological sequelae [1,2]. In addition, HIE has been implicated in further cognitive, motor, behavior, and/or language impairments, which account for approximately 20% of life-long disabilities, such as cerebral palsy [3]. One avenue of treatment is early detection, which allows for timely intervention and guidance for determining the type of therapy needed, such as therapeutic hyperthermia (TH). Currently, magnetic resonance imaging (MRI) allows the observation of primary structural brain injury patterns and secondary sequelae in HIE patients, which is the standard of care imaging choice. Specifically, MRI can noninvasively detect perinatally acquired cerebral lesions associated with HIE and characterize lesion severity [4,5,6,7], which can be used to prognosticate patients and provide treatment [8,9]. Therefore, providing accurate characterization of high-risk neonates is a necessary and important goal to improve therapy and potentially minimize poor neurological sequelae.

The MRI National Institute of Child Health and Human Development (NICHD) scoring system is presently the most commonly used for HIE evaluation method and has been shown to be a predictor for death and disability at 18–22 months following TH for HIE infants [10]. The NICHD score assesses the presence of lesions in the basal ganglia, thalamus, and posterior limb of the internal capsule (PLIC) [11], which have been shown to be predictors of abnormal motor outcomes [12]. For example, Miller et al. [6] observed that HIE patients with lesions in the basal ganglia and thalamus demonstrate the greatest impairment of motor and cognitive outcomes associated with cerebral palsy at 30 months. In addition, brain MRI examinations were shown to be a predictor of long-term neurological outcomes [6,12]. While such scoring systems are useful, potential limitations of these approaches are (1) the requirement for subjective visual inspection of abnormalities on MRIs and (2) the assignment into one of six existing scores, which is arbitrary. As a result, HIE patients without lesions involving the basal ganglia, thalamus, or PLIC may go on to have a long-term disability (false negatives), which may result in undertreatment. For example, Natarajan et al. observed that 17% of patients with pattern 1A and 25% of patients with pattern 1B developed disability/mortality [13]. Shankaran et al. also observed that HIE patients with grade 1b lesion of NICHD score still had 25% disability/mortality [14]. Conversely, some HIE patients with lesions involving the deep gray nuclei may go on to experience no adverse events (false positives), which may result in unnecessary treatment and follow-up tests. For example, Rutherford observed that up to 50% of HIE patients with a high NICHD score were not disabled at 18 months of age [15].

One further limitation of the NICHD scoring system is the lack of objective and quantifiable information. For example, the scoring system does not account for the (1) number of lesions or (2) size of lesions, which may better reflect the overall disease burden. Within other neurologic conditions, both size and lesion count have been shown to be important for prognostication. For example, an infarct volume of >75–100 mL has a higher risk of poor outcome in adult ischemic stroke patients [16]. Likewise, the increases in the volumes of the lesions seen on MRIs of multiple sclerosis patients correlate with the degree of long-term disability [17]. Therefore, it is possible that there is a similar relationship between disability and HIE lesion counts and volume for HIE patients.

The purpose of this study is to evaluate the distribution and burden of MRI changes as a prognostic indicator of neurodevelopmental (ND) outcomes at 18–24 months in HIE infants.

## 2. Materials and Methods

### 2.1. Subject Selection

We retrospectively identified a cohort of HIE patients between June 2013 and March 2017. Inclusion criteria were gestational age ≥35 weeks with a birth weight of ≥2000 g who underwent therapeutic hypothermia (TH) within 6 h of life. All patients received the same TH protocol, which is detailed in Appendix A. Exclusion criteria were HIE infants who were older than 6 h at the time of the assessment or those with other major congenital abnormalities, syndromes, or metabolic diseases. Infants with a birth weight of ≤2000 g, a gestational age of ≤35 weeks of gestational age, overt bleeding, signs of infection, or those requiring ≥80% oxygen support, which may suggest persistent pulmonary hypertension, were also excluded from this study. The perinatal history and delivery events were recorded for all patients.

#### Secondary Outcome (Follow-Up at 18–24 Months)

At 18~24 months, 107 infants returned for follow-up evaluations, at which time they completed the cognitive, language, and motor composites of the Bayley Scales of Infant and Toddler Development III and were evaluated by certified examiners (Figure 1). Children were considered as having developmental delay (DD) or abnormal development if their score was <85. If the score was ≥85, their ND stage was normal. The study was approved by the Ethics Committee of Seoul St. Mary’s Hospital, The Catholic University of Seoul, Korea (KC19REGI0095).

### 2.2. Imaging Methods

Brain MRI with diffusion was performed on all TH-treated infants (within at least 10 days of life) after they had been rewarmed and extubated from the ventilator. For each infant, diffusion weighted brain image (DWI) sequences from a 3T Siemens scanner were obtained for analysis. DWIs were analyzed both qualitatively and quantitatively for the extent and pattern of brain injury. For qualitative assessment, all images were independently reviewed by two neuroradiologists who were blinded to the final outcome. Additionally, for quantitative assessment, both reviewers created manual 3D segmentation masks for all DWI lesions. Based on these annotations, quantitative metrics including lesion count, size, and distribution were extracted. DWI sequences were assessed for each patient with the following parameters: TR 5100 ms, TE 99 ms, flip angle 90°, FOV 149 mm, matrix 256 × 256, and section thickness 4 mm.

### 2.3. Image Assessment

For qualitative assessment, all images were independently reviewed by two neuroradiologists (SI and DC) who were blinded to the final outcome. Two observers independently extracted data and assessed data quality and validity.

MRIs were categorized based on the patterns of structural injury according to the NICHD pattern for brain injury: score of 0 for a normal MR image; 1A for only minimal cerebral lesions; 1B for more-extensive cerebral lesions without the basal ganglia and thalamus (BGT), or posterior limb of the internal capsule (PLIC), or anterior limb of internal capsule (ALIC) involvement and no area of watershed infarction; 2A for any BGT, PLIC, or ALIC involvement or watershed infarction but with no cerebral lesions; 2B for any BGT, PLIC, or ALIC involvement or watershed infarction with additional cerebral lesions; and 3 for cerebral hemispheric devastation [10] (Figure 1). For quantitative assessment, all MRIs were annotated by a radiologist (PC) who was blinded to clinical outcomes using a previously described web-based tool for neuroimaging.

### 2.4. Statistical Analysis

Continuous variables are expressed as the means ± standard deviation (SD) and were compared with Student’s *t*-test. Continuous variables are displayed as the median with the interquartile range when variables were not normally distributed, and they were compared using the Wilcoxon rank sum test.

All brain lesions that were present on abnormal scans were characterized by their volume size in pixels and total count for each MR image. An additional analysis was performed using more granular and detailed information about abnormal features directly obtained from MR images with programming tools. Brain lesion sizes and counts were characterized by continuous values calculated programmatically, and then divided into four size groups for the statistical analysis. Each size group contained lesions with sizes of 0~50, 50~200, 200~500, and >500 pixels. Each lesion was counted separately from the adjacent ones if they were spatially separated from one another without touching. The total number of lesions in each size group was counted. Normal scans had zero lesion counts for all groups.

All lesions calculations were performed with Python 3.5 using the scipy and numpy libraries to determine the presence of lesions in an image, and their counts and sizes. As a result, a table with the total count per size range per patient was generated as an input to determine correlations between abnormalities on brain MR images and ND abnormalities at 18~24 months of age. All inferential statistical analyses were 2-tailed, with statistical significance defined as values of *p* < 0.05. Statistical analyses were performed with SAS v9.4. Lesion feature calculations were performed using Python 3.5 programming language.

## 3. Results

### 3.1. Demographic Characteristics

The study recruited 120 TH-treated infants between 2012 and 2016. Of these infants, two died before further studies (e.g., brain images) were performed. Another 11 patients were excluded from this study because the diffusion weighted brain images (DWI) were not available for further analysis. As a result, 107 TH treated infants were enrolled in this study; 33.6% (36/107) infants had normal brain MRI and 66.3% (71/107) had abnormal MRI findings. At 18–24 months, all 107 infants were evaluated for the neurodevelopmental assessment of the Bayley Scales of Infant and Toddler Development III by certified examiners; 31 of 107 infants (30.0%) had delayed neurodevelopment and 76 of 107 infants (70.0%) were normal in their neurodevelopmental (ND) stages, as shown in Figure 1.

Descriptive clinical characteristics of TH-treated infants are presented in Table 1. Generally, clinical characteristics were similar between the normal and abnormal ND groups, other than clinical seizures (*p* < 0.001), which were significantly higher in the abnormal ND group (Table 1).

### 3.2. MRI NICHD Scoring System for ND Outcome at 18–24 Months

At 18–24 months, 70.0% (76/107) of infants showed normal ND stages, and 30.0% (31/107) exhibited abnormal ND stages. According to MRI NICHD Scoring System, more lesions involved in the basal ganglia and thalamus, or posterior limb of internal capsule as grouped in stages 2A and 2B, were significantly associated with abnormal ND outcomes at 18~24 months of age (*p* < 0.001). Ten (32.26%) infants in the abnormal ND group were in stage 2 A, involving BGT, PLIC, or ALIC involvement or watershed infarction without additional cerebral lesions. The stage 2B was only found in the abnormal ND group. The most severe stage for cerebral hemispheric devastation, stage 3, was not found in either group in this study (Table 2).

### 3.3. Total Lesion Count and Size in the Brain MRI in Association with ND Outcome at 18–24 Months

Among the 107 HIE patients with TH, 8.3% (3/36) patients from the normal ND group did not have lesions in the brain MRIs. The mean size of lesion was significantly larger in the abnormal ND group (384.77 vs. 91.25) (*p* < 0.001). Similarly, the total lesion count was also significantly larger in the abnormal ND group (14.16 vs. 5.29). The subgroups with counts <2 and between 14 and 40 were significantly more related to the abnormal ND group. The sizes of lesions, such as the 0–100, 100–200 and greater than 500 subgroups, were significantly associated with the ND group (Table 3).

### 3.4. Receiver Operating Characteristic (ROC) Curve

The ROC curve comparing the predictive ability of HIE size/count manifested that the strongest predictor of ND outcome was the NICHD scoring system (Figure 2). The MRI NICHD showed AUC of 0.756, while the lesion size AUC was 0.718. The lesion count AUC was 0.705.

### 3.5. Interreader Reliability

After all images were independently reviewed by two neuroradiologists (SI and DC) who were blinded to the final outcomes of HIE infants, MRIs were categorized based on patterns of structural injury according to the NICHD scoring system. The Pearson correlation and inter-reader reliability between the readers were 0.683 and 0.567, respectively.

## 4. Discussion

In this study, we sought to observe the relationship between lesion volume and count in brain MRIs and patterns of HIE lesions in MRI images associated with ND outcomes at 18~24 months of age. This study also compares the predictive ability of HIE size/count against the NICHD scoring system. We observed that greater involvement of total lesion count, increased size and greater extension of injury in areas of the basal ganglia and thalamus, and a trend towards more abnormal scans in brain MRI were significantly associated with abnormal ND at 18–24 months of age, which was in line with our initial hypothesis. The additional statistical analysis observing the volumetric brain lesion count in MRI, in addition to studying the location of injury according to MRI NICHD Score systems, allowed a better characterization of brain injury sizes and patterns from neonatal brain MRI and improved prediction of the neurodevelopmental outcome. The lack of objective and quantifiable information in the NICHD scoring system was further supported by the objective counts of lesion size; the lesion size > 500 was significantly more often found in the abnormal ND outcome.

MRI is a noninvasive method and can be an optimum technique to assess perinatally acquired cerebral lesions associated with HIE [4,7], and a study of patterns and severity of lesions provides a guide for detecting adverse outcomes [10,12,18]. The correlation between the injury location and the development of infantile spasms [19] was also studied, which emphasized the importance of the injury location. Severe acute hypoxic-ischemic insults and lesions in the basal ganglia and thalamus are often associated with abnormalities [11]. In our study, an abnormal signal intensity in the basal ganglia and thalamus was a powerful predictor of abnormal ND outcomes [12] at 18~24 months of age (Table 2). Lesions in the basal ganglia and thalamus are usually consistent with a severe acute hypoxic-ischemic insult and often manifest with abnormalities in the cortex and adjacent subcortical white matter, which may be graded as a severe acute hypoxic-ischemic insult [8]. Moderate and severe lesions in the basal ganglia and thalamus and severe white matter lesions are associated with cerebral palsy [20]. One limitation of the NICHD scoring system is the lack of objective and quantifiable information. For example, the scoring system does not account for the (1) number of lesions or (2) size of lesions, which may better reflect overall disease burden. Within other neurologic conditions, these factors have been important in prognostication. For example, traumatic brain injury and diseases related to stroke or hemorrhage volume affected the prognosis. The study on the reliability of measuring regional brain volumes in infants with HIE showed that selected volumetric MRI findings in HIE infants correlated with neurosensory impairments at 18–22 months of age [21].

Although brain MRI examinations in locations of the brain were shown to be a predictor of long-term neurological outcomes [6,14], HIE patients without lesions involving the basal ganglia, thalamus, or PLIC may go on to experience long-term disability (false negatives). For example, Barnett et al. observed a 27% disability/impairment rate in neonates with normal MRIs [22]. Shankaran et al. also observed that among the 50 children who had a normal MRI scans, 30% manifested mild disabilities and 13% had moderate disability in a long-term prognosis [14].

Accordingly, there are studies that, within HIE patients, selected volumetric MRI findings correlated with neurosensory impairments, and total brain volumes correlated significantly with death or neurosensory impairment [21]. More studies demonstrated that a volumetric brain MRI study is a feasible and reliable surrogate measure predictive of long-term outcomes. A higher volume of acute brain injury involved was associated with worse 12-month neurodevelopmental outcomes [23]. Markus et al. reported that these volumetric measures were correlated with neurological outcome [24]. Andronikou et al. demonstrated regional cortical volume loss in mild and severe partial-prolonged hypoxic ischemic injury [25]. The characteristic study volume features in brain MRI of HIE can be of value in the communication of the nature and severity of HIE insult. Those volume assays can be handled and viewed in more detail in future studies. This study had some limitations. Several factors may have contributed to a potential selection bias in our review: first, this was a retrospective study design, which might be unable to fully confirm the examined relationships; second, we only had a relatively small sample size in study group; third, hidden disabilities may subsequently have become apparent, and many infants might have important developmental lags that were not classified as impairments. Despite the limitations, automation and the scanning speed of detecting abnormalities combined with an analysis of lesion volume and counts can assist clinicians in predicting disabilities in infants in the future.

## 5. Conclusions

An MRI assessment of the location and size/count of injury patterns can be a valuable predictor of ND outcomes in neonatal HIE infants. With greater involvement of the basal ganglia and thalamus and a trend towards more abnormal scans, a worse prognosis can be prepared for in advance [23]. In future studies, examination of longer-term data of HIE infants to assess unreported milder disabilities such as attention deficit disorders in the school-aged years is also warranted.

## Figures and Tables

**Figure 1 brainsci-10-00991-f001:**
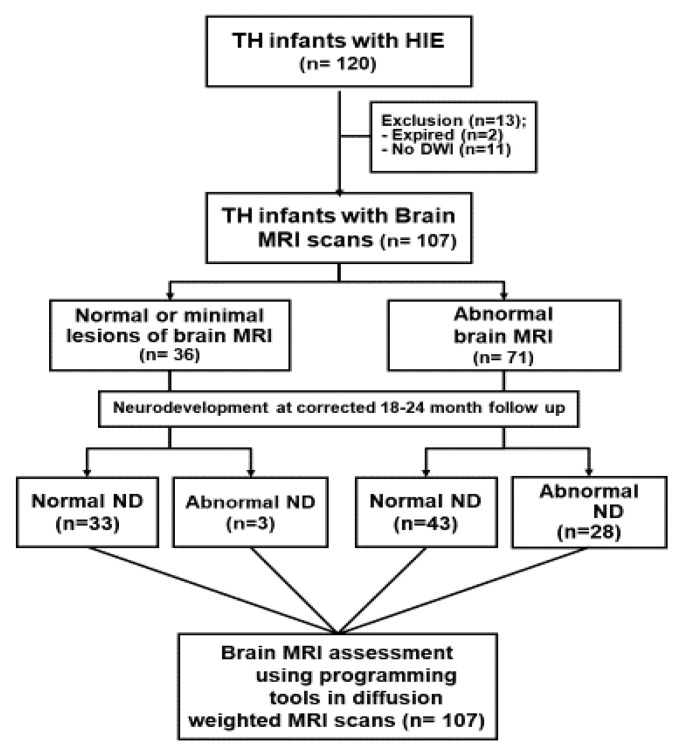
Flow chart of study. Abbreviations: TH, therapeutic hypothermia; HIE, hypoxic–ischemic encephalopathy; MRI, magnetic resonance imaging; ND; neurodevelopment, AI; artificial intelligence.

**Figure 2 brainsci-10-00991-f002:**
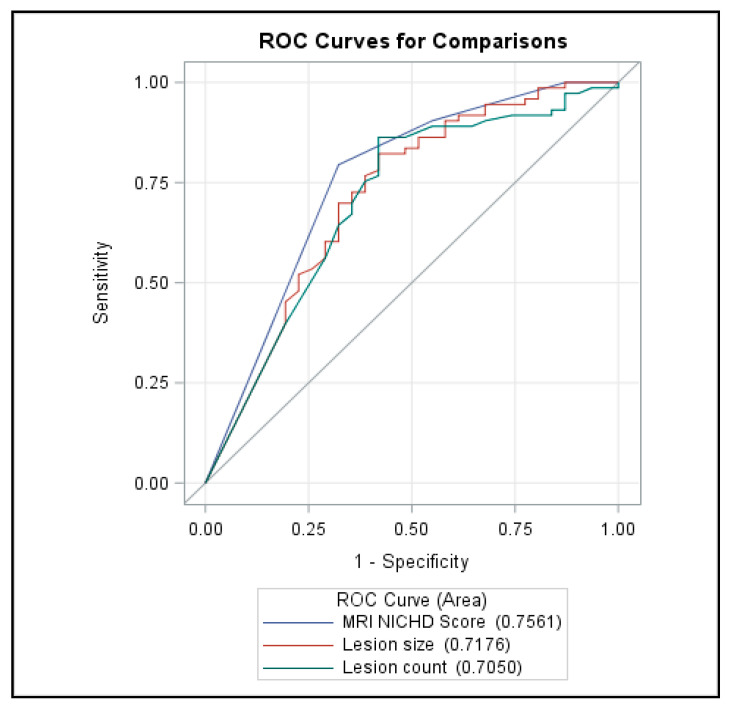
Receiver Operating Characteristic (ROC) curves for comparisons between MRI National Institute of Child Health and Human Development (NICHD) score (location) and lesion size and count.

**Table 1 brainsci-10-00991-t001:** Clinical variables related to neurodevelopmental outcomes at corrected 18~24 months of age (*n* = 107).

	Normal Neurodevelopment (*n* = 76)	Abnormal Neurodevelopment (*n* = 31)	*p*-Value
Gestational age, weeks	39.9 (38.8~40.2)	39.3 (38.2~40.0)	0.072
Birth weight, g	3230 (3030~3510)	3200 (2955~3370)	0.153
Delivery mode, emergent C/S	22 (26.5)	9 (25.7)	0.113
Apgar score at 1 min	4.90 (2.39)	4.69 (2.03)	0.638
Apgar score at 5 min	6.81 (2.01)	6.71 (1.87)	0.815
Male, *n* (%)	35 (42.2)	11 (31.4)	0.376
Sarnat Stage 1, mild	11 (14.5)	0 (0)	0.471
Sarnat Stage 2, moderate	54 (71.0)	26 (83.9)	
Sarnat Stage 3, severe	11 (14.5)	5 (16.1)	
Clinical seizure	55 (73.3)	31 (96.9)	0.011
Ventilator care, days	3.86 (4.82)	3.66 (2.06)	0.822
Full feeding (100 cc/kg/day)	8.53 (5.28)	8.18 (3.14)	0.724
Hospital days	13.80 (7.62)	13.54 (4.82)	0.857
Early intervention	22 (28.9)	20 (64.5)	0.002

**Table 2 brainsci-10-00991-t002:** Magnetic resonance imaging (MRI) findings according to the MRI NICHD Score (*n* = 107).

MRI NICHD Score, *n* (%)	Normal Neurodevelopment (*n* = 76)	Abnormal Neurodevelopment (*n* = 31)	*p*-Value
0	30 (39.47)	5 (16.13)	<0.001
1A	31 (40.79)	5 (16.13)	<0.001
1B	8 (10.53)	7 (22.58)	<0.001
2A	7 (9.21)	10 (32.26)	<0.001
2B	0 (0)	4 (12.90)	<0.001
3	0 (0)	0 (0)	NA

MRI were classified according to the NICHD pattern for brain injury: score of 0 for normal MRI; 1A for minimal cerebral lesions only; 1B for more extensive cerebral lesions without the basal ganglia and thalamus (BGT), or posterior limb of internal capsule (PLIC) or anterior limb of internal capsule (ALIC) involvement and no area of watershed infarction; 2A for any BGT, PLIC, or ALIC involvement or watershed infarction without any cerebral lesions; 2B for any BGT, PLIC, or ALIC involvement or watershed infarction with additional cerebral lesions; and 3 for cerebral hemispheric devastation.

**Table 3 brainsci-10-00991-t003:** Total lesion count and size in the brain MRI in association with neurodevelopmental (ND) outcomes at 18–24 months.

Brain MRI	Normal ND (*n* = 76)	Abnormal ND (*n* = 31)	*p*-Value
size (mean (SD))	91.25 (179.63)	384.77 (527.18)	<0.001
lesion size < 100	59 (77.6)	12 (38.7)	<0.001
lesion size 100–200	5 (6.8)	3 (9.7)	0.926
lesion size 200–500	8 (11)	7 (22.6)	0.216
lesion *n* size > 500	4 (5.5)	9 (29)	<0.001
Total lesion count (mean (SD))	5.29 (9.29)	14.16 (12.49)	<0.001
counts < 2	44 (57.9)	9 (29.03)	0.0113
count between 2–14	22 (30.14)	4 (12.90)	0.0634
count between 14–40	9 (12)	16 (51)	<0.001
count > 40	1 (1.37)	2 (6.45)	0.1567
Number of lesions with volume size between 0–100 (mean (SD))	5.29 (9.29)	14.16 (12.49)	<0.001
between 100–200 (mean (SD))	0.18 (0.54)	0.52(0.81)	0.014
between 200–500 (mean (SD))	0.07 (0.38)	0.10 (0.40)	0.735
above 500 (mean (SD))	0.00 (0.00)	0.06 (0.25)	0.028

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
