# Peer review of "Predictive Values of Location and Volumetric MRI Injury Patterns for Neurodevelopmental Outcomes in Hypoxic-Ischemic Encephalopathy Neonates"

_brainsci, 2020, doi:10.3390/brainsci10120991_

Round 1
Reviewer 1 Report
Congratulations for your excellent work, very innovative. For sure, MRI information to consider in our clinical practice and patient prognosis.
But there are some issues to comment:
- Cooling method.
In this retrospective study, two cooling methods were employed. Results comparing both therapies are diverse, but some of them point that full-body cooling would provide a broader area of protection (Allen K. Moderate Hypothermia: Is Selective Head Cooling or Whole Body Cooling Better? Adv Neonatal Care. 2014 Apr; 14(2): 113–118.)
Maybe results should be analyzed considering cooling therapy applied.
- Patients clinical HIE features
Detailed encephalopathy classification of patients is missing (number of patients on mild, moderated or severe HIE).
- Follow up
There is no description about follow up program.
-How many patients attended follow up?
-How many of them did not undergo?
-How many patients were lost to follow up?
-How many patients received intervention (early intervention)?
- Keeping in mind points 2 and 3, results in Normal/Abnormal ND according to Bayley III scale might be modified depending on: encephalopathy scale and follow up and intervention program.
Clinical information about these issues is needed.
- Fig 1-RMN classification: What is considered NORMAL OR MINIMAL LESIONS (stage 0,1A, 1B)? No information detailed.
- Page 5: There are some issues about Table 3
- 3.- First line: Among of 107 patients with TH, 8,3% (3/36) patients form the abnormal ND group… Table 3, Abnormal ND group is 36 patients.
- Results of total count lesion addition does not match with total patients in both groups (73 vs 71 patients in normal ND, 34 vs 36 in abnormal ND)
- Page 7: many clinical conditions have commingled. Could you please explain this comment?
Author Response
Congratulations for your excellent work, very innovative. For sure, MRI information to consider in our clinical practice and patient prognosis.
But there are some issues to comment:
Thank you so much for reviewing our article and proving your valuable comments
- Cooling method.
In this retrospective study, two cooling methods were employed. Results comparing both therapies are diverse, but some of them point that full-body cooling would provide a broader area of protection (Allen K. Moderate Hypothermia: Is Selective Head Cooling or Whole Body Cooling Better? Adv Neonatal Care. 2014 Apr; 14(2): 113–118.)
Maybe results should be analyzed considering cooling therapy applied.
Out of 107 patients, only 3 patients received selective head cooling. The rest 104 (97%) patients received whole Body Cooling. For these reasons, the two cooling therapy methods can not be analyzed.
- Patients clinical HIE features
Detailed encephalopathy classification of patients is missing (number of patients on mild, moderated or severe HIE).
The HIE stages are inserted as you suggested in Table 1 as below;
|
Sarnat Stage 1, mild |
11 (14.5) |
0 (0) |
0.471 |
|
Sarnat Stage 2, moderate |
54 (71.0) |
26 (83.9) |
|
|
Sarnat Stage 3, severe |
11 (14.5) |
5 (16.1) |
|
- Follow up
There is no description about follow up program.
We described more in detail of the study flow chart in 3.1. Demographic Characteristics LINE 144-152.
-How many patients attended follow up?
The study recruited 120 TH-treated infants between 2012 to 2016.
-How many of them did not undergo?
Of these infants, two died before further studies (e.g., brain images) were performed. Another 11 patients were excluded in this study because the diffusion weighted brain images (DWI) were not available for further analysis. A total 13 patients were excluded.
-How many patients were lost to follow up?
At 18-24 months, all 107 infants were evaluated for the neurodevelopmental assessment of the Bayley Scales of Infant and Toddler Development III by certified examiners. No one was lost to follow up.
-How many patients received intervention (early intervention)?
We added this in Table 1 as below;
|
Early intervention |
22 (28.9) |
20 (64.5) |
0.002 |
Also, we described more in detail of the study flow chart in 3.1. Demographic Characteristics in LINE 144-152; “The study recruited 120 TH-treated infants between 2012 to 2016. Of these infants, two died before further studies (e.g., brain images) were performed. Another 11 patients were excluded in this study because the diffusion weighted brain images (DWI) were not available for further analysis. As a result, 107 TH treated infants were enrolled in this study; 33.6% (36/107) infants had normal brain MRI and 66.3% (71/107) had abnormal MRI findings. At 18-24 months, all 107 infants were evaluated for the neurodevelopmental assessment of the Bayley Scales of Infant and Toddler Development III by certified examiners; 31 of 107 infants (30.0%) had delayed neurodevelopment and 76 of 107 infants (70.0%) were normal in their neurodevelopmental (ND) stages as in Figure 1.”
- Keeping in mind points 2 and 3, results in Normal/Abnormal ND according to Bayley III scale might be modified depending on: encephalopathy scale and follow up and intervention program.
Clinical information about these issues is needed.
We added those information in Table 1 as you suggested.
- Fig 1-RMN classification: What is considered NORMAL OR MINIMAL LESIONS (stage 0,1A, 1B)? No information detailed.
The normal finding was when the MRI was clearly normal as in score 0 in NICHD scoring system and abnormal findings included any lesions from minor cerebral lesions to deep brain injuries as in scores from 1 A to 3 in NICHD scoring system.
- Page 5: There are some issues about Table 3
- 3.- First line: Among of 107 patients with TH, 8,3% (3/36) patients form the abnormal ND group… Table 3, Abnormal ND group is 36 patients.
- Results of total count lesion addition does not match with total patients in both groups (73 vs 71 patients in normal ND, 34 vs 36 in abnormal ND)
Thank you for pointing out the number. The number in the lesion size was corrected. The 76 normal ND and 31 abnormal ND groups are all corrected in numbers as below in Table 3.
|
Brain MRI |
Normal ND (n=76) |
Abnormal ND (n=31) |
p-value |
|
lesion size <100 |
59(77.6) |
12(38.7) |
<0.001 |
|
counts <2 |
44(57.9) |
9(29.03) |
0.0113 |
- Page 7: many clinical conditions have commingled. Could you please explain this comment?
We decided to take out the third limitation. Thank you again for your valuable comments.!

Reviewer 2 Report
The present study examines specific lesion characteristics in HIE patients after TH, in addition to scoring using the NICHD system. The authors highlight the limitations of the currently favoured NICHD scoring system, explaining the long-term consequences of missed or inappropriate treatment that false positives or false negatives might provide. The authors also emphasise the size and quantity of lesions as important factors that are not taken into consideration with the NICHD score, which have been shown in other conditions to predict long-term disability. The manuscript is well-written, with a couple of things requiring clarification as follows:
- Authors should clarify in the inclusion criteria that TH needed to be initiated within 6h of birth (and that it was not MRI acquired in this timeframe).
- More information is needed on the DWI acquisition protocol (and potentially, move it to the top of the section).
- When the authors first describe “abnormal findings” in the results, it should be clarified as to whether these are ‘abnormal’ per NICHD criteria or by the author’s more extensive measures.
- There is repetition of result detail in sections 3.1 and 3.2 – this should be tightened up.
- Line 46: A word seems to be missing, or a restructure is needed of that sentence to describe it as a scoring system for MRI data.
- Line 73: typo – “evaluate the”.
- Repetition on ethics approval first and last sentences of 2.1. Subject Selection
- Line 160: revise sentence “..the most shown as in..”
Author Response
The present study examines specific lesion characteristics in HIE patients after TH, in addition to scoring using the NICHD system. The authors highlight the limitations of the currently favoured NICHD scoring system, explaining the long-term consequences of missed or inappropriate treatment that false positives or false negatives might provide. The authors also emphasise the size and quantity of lesions as important factors that are not taken into consideration with the NICHD score, which have been shown in other conditions to predict long-term disability. The manuscript is well-written, with a couple of things requiring clarification as follows:
Thank you so much for reviewing our article and proving your valuable comments.
- Authors should clarify in the inclusion criteria that TH needed to be initiated within 6h of birth (and that it was not MRI acquired in this timeframe).
To clarify, In LINE 81, we added “within 6 hours of life” in the inclusion criteria of TH. LINE 78-79; “Inclusion criteria were gestational age ≥ 35 weeks with a birth weight of ≥2000 g who underwent therapeutic hypothermia (TH) within 6 hours of life.”
Also, we moved the sentence in LINE 101, “Brain MRI with diffusion was performed on all TH-treated infants (within at least 10 days of life) after they had been rewarmed and extubated from the ventilator.” to LINE 94 to clarify.
- More information is needed on the DWI acquisition protocol (and potentially, move it to the top of the section).
We moved the sentence to the top of the section as you recommended.
LINE 92-94; “Brain MRI with diffusion was performed on all TH-treated infants (within at least 10 days of life) after they had been rewarmed and extubated from the ventilator. For each infant, DWI sequences from a 3T Siemens scanner were obtained for analysis.”
- When the authors first describe “abnormal findings” in the results, it should be clarified as to whether these are ‘abnormal’ per NICHD criteria or by the author’s more extensive measures.
The normal finding was when the MRI was clearly normal as in score 0 in NICHD scoring system and abnormal findings included any lesions from minor cerebral lesions to deep brain injuries as in scores from 1 A to 3 in NICHD scoring system.
We also corrected “MRI” to “neurodevelopmental groups” in LINE 150-151; “Generally, clinical characteristics were similar between the normal and abnormal ND groups other than clinical seizures (p<0.001) which were significantly higher in the abnormal ND group (Table 1).”
- There is repetition of result detail in sections 3.1 and 3.2 – this should be tightened up.
Thank you for your comment. We deleted the redundant sentence in the beginning of the parahraph and changed as below;
Line 156-159;“At 18-24 months, 70.0% (76/107) of infants showed normal ND stages, 30.0% (31/107) exhibited abnormal ND stages. According to MRI NICHD Scoring System, more lesions involved in the basal ganglia and thalamus, or posterior limb of internal capsule as grouped in stages 2A and 2B, were significantly associated with abnormal ND outcomes at 18~24 months of age (p<0.001).”
- LINE 46;A word seems to be missing, or a restructure is needed of that sentence to describe it as a scoring system for MRI data.
The sentence was reparagraphed as below;
LINE 46; “The MRI National Institute of Child Health and Human Development (NICHD) scoring system is presently the most commonly used for HIE evaluation and shown to be a predictor for death and disability at 18–22 months following TH for HIE infants [10].”
- Line 73: typo – “evaluate the”.
Thank you. We corrected as the following;
LINE 73-74; “The purpose of this study is to evaluate the distribution and burden of MRI changes as a prognostic indicator of neurodevelopmental (ND) outcomes at 18-24 months in HIE infants.”
- Repetition on ethics approval first and last sentences of 2.1. Subject Selection
Thank you. We deleted the first sentence for the redundancy.
- Line 160: revise sentence “..the most shown as in..”
We removed the “..the most shown as in..”
Once again, thank you so much for your valuable comments.